# MixedSAND: Semantic Annotation of Mixed-unit Numeric Data

| Entity | C1 | C2 | Source |
|---|---|---|---|
| **Player** | 1.88 | 84 | espn.co.uk |
| | 1.85 | 87 | |
| | 1.85 | 93 | |
| | 1.93 | 104 | |
| | 6.25 | 192 | nhl.com |
| | 6.17 | 218 | |
| | 6.25 | 208 | |

Table 1: The height and weight of NHL players integrated from two sources: https://www.espn.co.uk/ and https://records.nhl.com/.

## ABSTRACT

Quantitative information about entities constitutes a significant portion of tabular data in open sources and data lakes. Such tables often lack consistent labeling and proper schema, posing significant challenges for querying and integration. This paper studies the problem of numerical column annotation in scenarios where quantitative data may be gathered from different sources and unit consistency is a concern. For instance, weight measurements may vary between entities, expressed in kilograms for some and pounds for others, with no accompanying unit information. We investigate the conditions for effectively annotating mixed-unit numeric data, introduce a benchmark for such an annotation task, and propose an algorithm that reliably detects semantic types (e.g., height) and links them to the corresponding types present in a knowledge graph. Our evaluation on a diverse set of columns with mixed units and varying levels of annotation difficulty shows that our method significantly outperforms strong baselines such as GPT-4o-mini and SAND in terms of accuracy, excelling in both detecting mixed units and annotating them with appropriate semantic labels. (All our code and data will be publicly released upon acceptance of the paper.)

## CCS CONCEPTS

• **Information systems** → **Information integration**; **Data extraction and integration**; **Data mining**.

## KEYWORDS

Column annotation, numeric data, semantic annotation

**ACM Reference Format:**
Anonymous Author(s). 2025. MixedSAND: Semantic Annotation of Mixed-unit Numeric Data. In *Proceedings of Proceedings of the ACM Web Conference (Web '25).* ACM, New York, NY, USA, 10 pages. https://doi.org/XXXXXXX.XXXXXXX

## 1 INTRODUCTION

An ever growing number of tables reside in data lakes and open government sources [14], and many of those tables contain highly valuable data for tasks such as question answering [15, 25], fact verification [6] and decision making processes [13]. To leverage those tables in downstream tasks, one needs to know the semantics of the columns and their relationships. Tables within corporate databases may also use semantic labels to boost the performance of query generation tools [19]. Quantitative information about entities,

in particular, constitutes a significant portion of columns in these table repositories, yet their semantics remain often inadequately represented. For example, consider the list of hockey players, shown in Table 1, sourced from two sporting websites and integrated into a single table without meticulous examination or mediation. One data source represents the height (column C1) in meters and the weight (column C2) in kilograms, while the other data source expresses the height in feet and the weight in pounds[1]. The issue of mixed-unit columns becomes particularly prominent in data integration tasks, such as table union and stitching[10, 12, 16]. For instance, during a table union operation, two columns representing the same attribute (e.g., height) may be combined despite being recorded in different units. Moreover, several data integration processes–such as data warehousing, ETL (Extract-Transform-Load), and the creation of unified data views–involve merging datasets from multiple sources, which can result in columns with mixed units. This amalgamation of data from disparate sources often overlooks unit consistency or conversion. To address this issue, we investigate whether numerical columns with mixed units can be identified and if accurate semantic labels can be assigned.

There are several challenges in mapping the columns of a table to a knowledge graph, and those challenges are magnified for numerical columns. Firstly, if the desired semantic type or column values do not exist in the knowledge graph, existing methods cannot make accurate predictions. Particularly with numerical columns, quantitative measurements of identical entities from different sources seldom match precisely, due to inherent inaccuracies in measurements and reporting. For example, the height of the Eiffel Tower is listed as 330 meters on Wikipedia[2], while it is stated as 324 meters on Wikidata[3]. Secondly, quantitative data frequently undergo changes over time due to the dynamic nature of measurements. For instance, while attributes such as name and nationality of a player are less likely to change, measurements such as height and weight can change regularly. Lastly, quantitative data may be reported using different units (e.g., kilograms and pounds for weight). Because of these issues, reliably determining the semantic type of

---

[1]In general, individual entity names may not be available for reasons such as privacy, and our approach does not assume that individual entity names are given.
[2]https://en.wikipedia.org/wiki/Eiffel_Tower
[3]https://www.wikidata.org/wiki/Q243

numerical data and assigning appropriate annotations still remains a challenging task.

Various methods exist for detecting the semantic types of columns, often involving their mapping to types in knowledge bases, but most of these approaches are tailored for textual data. Despite some attempts to adapt these methods for numerical data (e.g., [26, 27]), progress has been slow and the adaptability of the methods remain questionable. Many approaches targeting numerical data leverage the statistical properties of the columns, such as mean and standard deviation, along with statistical testing, to assign a type [9, 17, 18, 20]. However, the underlying assumptions of these tests, namely that the values of each semantic type follow a known distribution and that the query column is a random sample from the same distribution, are often violated. More recent work attempt to relax some of the assumptions on the distribution [24], yet all aforementioned works assume that column values are uniformly expressed using the same unit, which may not hold when data are gathered from multiple sources, as discussed earlier.

Our work aims to further relax the assumptions about data distribution. In particular, we do *not* make the assumption that column values are uniformly expressed using the same unit. In this context, our main contribution include a mixed-unit numeric annotation benchmark and a three-staged numeric data annotation pipeline. The pipeline consists of (1) model generation, where plausible models of data subsets are generated, and data points are assigned to those sub-models, (2) type annotation, where a semantic type is assigned to each sub-model, following a cost-based approach, and (3) an aggregation phase, where sub-model costs are aggregated to estimate the cost of each unifying model covering the entire column values, and to select models with the least cost. Our experimental evaluation on a diverse collection of data, including our benchmark and other datasets, demonstrates the superiority of our approach over strong baselines from the literature in both detecting and annotating mixed-unit numeric columns.

The remainder of this paper is organized as follows: Section 2 reviews related works, highlighting the limitations of existing methods. Section 3 details the construction of our datasets. Section 4 outlines our methodology, explaining the steps involved in the proposed approach. Section 5 presents the experimental setup and results, comparing our approach with the state-of-the-art methods. Finally, Section 7 concludes the paper, summarizing our contributions and discussing the implications of our findings and potential future work in this area.

## 2 RELATED WORK

### 2.1 Categorical and Textual Data Annotation

In recent years, significant progress has been made in the field of column type annotation for categorical and textual data. Various models and approaches have been developed to address the challenges associated with predicting column types in web tables. For instance, Chen et al. [5] proposed Colnet, a model that leverages the semantics of web tables to predict column types through embeddings, demonstrating a robust approach to semantic table interpretation. Efthymiou et al. [8] explored the matching of web tables with knowledge base entities, transitioning from simple entity lookups to more sophisticated entity embeddings, thus enhancing

the accuracy of semantic annotations. Zhang's TableMiner [28] is another notable contribution, providing an effective and efficient method for semantic table interpretation by integrating multiple sources of evidence for type prediction. Zhang et al. [27] introduced Sato, a contextual semantic type detection system that utilizes context to improve the prediction accuracy of column types in tables. Finally, the SemTab Challenge [1], held for the past five consecutive years, is a benchmark for mapping tabular data to a Knowledge graph with a primary focus on textual data.

### 2.2 Numerical Data Annotation

While substantial advancements have been made for categorical and textual data, the task of annotating numerical columns has also seen progress, though with some limitations. SAND (Semantic Annotation of Numerical Data) [24] is a pioneering work in this domain, focusing on numerical data and outperforming traditional statistical-based methods such as KS-test. (We provide more details on SAND in Section 4.1.) Alobaid's work on fuzzy semantic labeling of semi-structured numerical datasets [3] introduces a novel approach to handle the inherent uncertainty in numerical data annotation. Neumaier's research on multi-level semantic labeling [17] provides a hierarchical approach to enhance the granularity and accuracy of numerical data annotations. Pham's domain-independent approach to semantic labeling [18] aims to generalize the annotation process across various domains, making it more versatile. Ramnandan's work on assigning semantic labels to data sources [20] contributes to improving the interoperability and usability of numerical datasets. Kacprzak's study on making sense of numerical data focuses on the semantic labeling of web tables [9], addressing the challenges of interpreting numerical columns in a web context.

Despite these advancements, existing methods for numerical data annotation often overlook the complexities, discussed in the previous section, introduced by columns containing multiple units of measurement. This limitation hampers the effectiveness of semantic annotations in such cases.

## 3 DATASET CONSTRUCTION

There is a lack of datasets for evaluating semantic annotation of numeric columns. The SemTab challenge [1], held annually since 2019, evaluates tabular data mapping to a knowledge graph, but it primarily includes textual columns. In SemTab Challenge 2021 [7], a task was introduced in Round 2 to identify the semantic relationship between an entity and a numeric property (e.g., ⟨Kielzugvogel,5.8⟩ and ⟨MT explosive motorboat,5.62⟩). However, this task differs from annotating a numerical column. Recent papers [3, 24] have introduced small manually annotated datasets, but all column values in these datasets have the same semantic type and unit. To our knowledge, there is no public multi-unit numeric column annotation dataset.

To fill this gap, we introduce a mixed-unit numeric column annotation dataset with varying levels of separability difficulty between units. To quantify this separability difficulty, we introduce the concept of *reflectivity* before discussing our dataset.

## 3.1 Reflectivity

Some mixed-unit columns are challenging to separate, even for human experts, due to significant overlap in data ranges from different units. In contrast, other mixed-unit scenarios are readily identifiable. To quantify this difficulty, we introduce the concept of *reflectivity*, which was previously used in a different context to quantify the interference between data dimensions [2].

Reflectivity, as defined by Agrawal and Srikant [2], measures the likelihood that a data point's reflection exists within the dataset. If $\vec{x}_i$ denotes the coordinates of a data point, the reflections of $\vec{x}_i$ include all permutations of $\vec{x}_i$'s coordinates, including $\vec{x}_i$ itself. For example, the reflections of (1,2) is {(1, 2), (2, 1)}. The reflectivity of a set of points $D \subseteq \mathbb{R}^2$ is formally defined as:

$$\text{Reflectivity}(D, r) = 1 - \frac{1}{|D|} \sum_{\vec{x}_i \in D} \frac{\theta(\vec{x}_i)}{\rho(\vec{x}_i)} \tag{1}$$

where $r$ is a distance threshold chosen experimentally, $\theta(\vec{x}_i)$ denotes the number of points within Euclidean distance $r$ of $\vec{x}_i$, $\rho(\vec{x}_i)$ is the number of points in $D$ with at least one reflection within distance $r$ of $\vec{x}_i$, $|D|$ is the dataset cardinality. The reflection of a point includes itself, hence $\rho(\vec{x}_i) \geq 1$ and $\rho(\vec{x}_i) \geq \theta(\vec{x}_i)$. Reflectivity is zero if $\rho(\vec{x}_i) = \theta(\vec{x}_i)$ for all data points. For higher dimensions, the reflectivity is computed as the average reflectivity across all 2-dimensional subspaces.

In our work, we are interested in the reflectivity relationship between different units of a property. Given two sets of quantities, $U_1$ and $U_2$, both measuring the same property but in different units, the values of $U_1$ can interfere with annotating the values in $U_2$ if their values are close. To capture this relationship between units, we create our dataset $D$ as the Cartesian product $U_1 \times U_2$, which includes every pair $(a, b)$ where $a \in U_1$ and $b \in U_2$. A high reflectivity in this dataset indicates a greater average number of reflections falling within distance $r$ of existing data points. Intuitively, this scenario implies an increased degree of overlap between the data represented by the units $U_1$ and $U_2$. This overlap makes it more challenging to separate these units. To illustrate, consider data points $(a_i, b_i)$, $(a_j, b_j) \in D$ that are not within distance $r$ of each other. If the reflection $(b_i, a_i)$ of $(a_i, b_i)$ is within distance $r$ of $(a_j, b_j)$, it suggests that the values $b_i$ and $a_j$, as well as $b_j$ and $a_i$, are similar. Since we know $a_i$ and $a_j$ belong to $U_1$, and $b_i$ and $b_j$ belong to $U_2$, this proximity of values from different units suggests a greater degree of overlap between the units. Conversely, a low reflectivity suggests that the values in different units are largely distinct, with minimal overlap. In our dataset generation, we set the value of $r$ such that the average number of points within distance $r$ is 2.5. This decision was made to capture reflections that fall within top 2-3 neighbours of a data point.

As an example of high reflectivity, consider soccer field sizes measured in meters and yards. As shown in Figure 1, many data points are reflected in close proximity to others, making it challenging to distinguish between the two units. In contrast, for an example of low reflectivity, consider player weight in pounds and kilograms. As illustrated in Figure 2, the units are highly separable, with reflections placed far from the original data points, making differentiation much easier.

| Entity | Size (meters) | Size (yards) |
|---|---|---|
| **Soccer Fields** | 100 | |
| | 84 | 91.86 |
| | 110 | |
| | | 100.58 |
| | 103 | 112.64 |
| | | 82.30 |

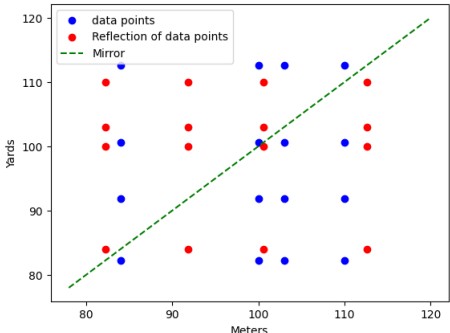

**Figure 1: Distribution of high reflective data (reflectivity = 0.875)**

| Entity | Weight (lb) | Weight (kg) |
|---|---|---|
| **Players** | 210 | 95.25 |
| | 194 | |
| | 150 | |
| | 176 | |
| | | 67 |
| | 178.56 | 81 |
| | | 73 |
| | | 92 |

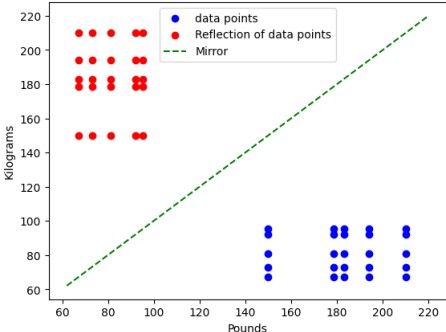

**Figure 2: Distribution of non-reflective data (reflectivity = 0)**

## 3.2 Mixed-Unit Dataset

Our Mixed-Unit dataset leverages real-world numeric data from Wikidata, transforming it into a robust dataset designed to test a model's ability to handle mixed units. The dataset is generated systematically, allowing control over key parameters such as unit counts, percentages of each unit within a column, and levels of data reflectivity within each column.

We generate three variations of the Mixed-unit dataset, each consisting of 200 columns with 30 rows per column. Each column consists of data in two different units, with an equal distribution of values from each unit. The values of different units in each column are present in varying proportions and degrees of overlap.

- Easy Dataset: This dataset is designed to present the least challenge in identifying mixed-unit columns. The majority of columns (60%) exhibit low reflectivity (< 0.3), indicating minimal overlap between the two units present within each of these columns. This characteristic makes the identification of mixed-unit columns relatively straightforward in this dataset. The remaining columns are divided equally between those with moderate reflectivity (0.3 - 0.6) and those with higher reflectivity (> 0.6).
- Medium Dataset: This dataset introduces a moderate level of difficulty in identifying mixed-unit columns. Compared to the Easy dataset, the majority of columns (60%) exhibit moderate reflectivity (0.3 - 0.6), indicating a noticeable overlap between data points with different units. The remaining columns are divided equally between those with low reflectivity (< 0.3) and those with higher reflectivity (> 0.6).
- Hard Dataset: This dataset presents the most challenging scenario for mixed-unit column identification. The majority of columns (60%) exhibit high reflectivity (> 0.8), making it significantly more difficult to distinguish between the two units and identify mixed-unit columns. The remaining columns are equally divided between those with low reflectivity (< 0.3) and those with moderate reflectivity (0.3 - 0.6).

## 4 MIXEDSAND APPROACH

Consider a table with a set of columns $c_1, \ldots, c_n$, and let val($c_i$) denote the set of values in column $c_i$. Our focus is on columns that consist of only numeric values. Each table often describes a set of entities (e.g., person, organization, location) or relationships between entities (e.g., distance between two cities), and this limits the set of types a column can take.

PROBLEM 1 (NUMERIC COLUMN TYPE ANNOTATION). *Let T be a set of entity types, P be a set of properties and U be a set of units. A semantic type can be denoted as a triple <t, p, u> where t ∈ T, p ∈ P and u ∈ U. The problem of column type annotation for a numeric column $c_q$ is the task of assigning a semantic type to $c_q$.*

In this work, we use a knowledge graph (KG) to construct our candidate semantic types. However, candidate types may also be obtained from other sources, such as Wikipedia or a training set. For each candidate semantic type $c$, we require a sample set of data points val($c$) representing the distribution. We operate under *the closed world assumption*, meaning that the candidate set is considered complete. This assumption is commonly made in similar approaches on annotating tabular data [5, 8, 24, 28].

### 4.1 Background on Single Unit Column Annotation

A few methods have been developed under the assumption that the input column consists of a single unit [3, 24]. Our methodology is built around SAND [24], a method that has been shown to be the state-of-the-art performance for annotating single-unit numeric columns, and which is reviewed in the following.

SAND [24] compares a given numeric column, known as the query column, with all candidate columns in the knowledge graph. This comparative analysis is executed by constructing a complete bipartite graph between the query column and the candidate column. On one side, nodes represent the numbers in the query column, while on the other side, they correspond to the numbers in one of the candidate columns. Edges in this graph signify the numerical disparity between the nodes they connect. Each mapping of the query column to a candidate column is represented with a subgraph and is associated with a cost. This cost is defined as the sum of the edge weights in the mapping, with each weight giving the difference between two quantities.

It should be noted that the model operates under the assumption that all query columns are single-unit columns, and each query column is compared with candidate columns that feature a singular unit. When units are mixed within the query column, the model maps values across different units from a mixed-unit query column to single-unit candidate columns. Consequently, the model is prone to fail to produce satisfactory results when the query column consists of mixed units, as shown in our evaluation section.

Furthermore, this approach presumes that all values in the query column align with units available in the knowledge graph. If the data exists in a different unit within the knowledge base, SAND's predictive accuracy is compromised.

### 4.2 Annotating Mixed-Unit Columns

Assuming that the column to be annotated has mixed units [4], our approach consists of a three-staged pipeline, as illustrated in Figure 3: (1) *model generation*, where plausible models of data subsets are generated, and data points are assigned to such sub-models, (2) *sub-model annotation*, where a semantic type is assigned to each sub-model, following a cost optimization framework for numerical data similar to SAND [24], and (3) *aggregation phase*, where sub-model costs are aggregated to estimate the cost of each unifying model covering the entire column values as well as to select models with the least cost.

*4.2.1 **Model Generation**. Given a column with mixed units, there can be potentially many possible groupings of the column values, with the total number determined by the powerset of the column values. We hypothesize that *quantities or measurements with the same unit are more likely to be closer to each other than those with different units*. Based on this hypothesis, a clustering of the column values can place values with the same unit in the same cluster. As the simplest and most commonly used clustering method, k-means is an option. However, there are two key problems that need to be addressed. First, the scale of the numbers can vary significantly between units (e.g., millimeters and kilometer), making the absolute difference between two quantities less meaningful when they are of different units. Second, the number of units in a mixed column is often unknown, which complicates the determination of the appropriate number of clusters k. We initially assume the

---

[4]This assumption is relaxed in Section 4.3.

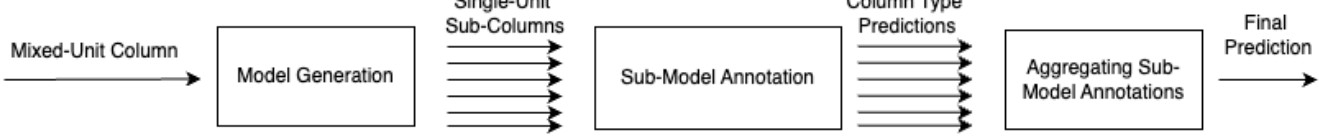

**Figure 3: MixedSAND column annotation pipeline**

number of units in a column is known and relax this condition in Section 4.3

**Choosing an appropriate distance function.** The default distance function in k-means is the Euclidean distance, which treats all differences between quantities equally. This poses a problem when clustering quantities with different measurement scales and units. For example, consider the "Area" column in Table 2, which lists the areas of Canadian cities in both square meters and square kilometers. Without knowing the units, the default k-means distance function (Euclidean distance) would incorrectly cluster the first three cities with areas 684.4, 115, and 87,430,000 together, while separating the fourth city with an area of 630,200,000. This error occurs because the default metric fails to account for the scale variations inherent in mixed-unit data, as depicted in Figure 4 using min-max normalization for clearer illustration.

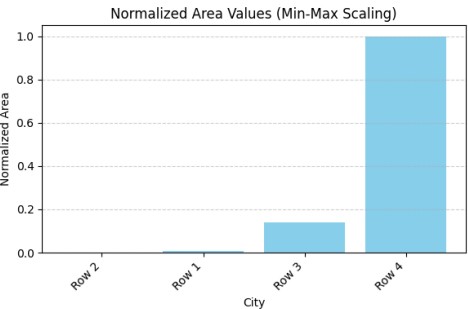

**Figure 4: Normalized area values (min-max scaling) for Canadian cities in Table 2. The visual gap between the third and fourth rows highlights the scaling issue leading to a misclassification.**

In contrast, humans typically have no problem recognizing these scale differences and will likely cluster the first two cities with measurements in square kilometers together and the last two cities with measurements in square meters together. Our proposed solution to this problem is the Bray-Curtis distance [21], a normalized relative difference function defined as follows:

$$\text{distance}(x, y) = \frac{|x - y|}{x + y} \quad \text{for } x, y \geq 0 \text{ and } x + y > 0. \quad (2)$$

This distance function, commonly used in ecology for comparing quantities with different scales, incorporates the magnitudes of quantities into the comparison, effectively replicating the intuitive clustering of city areas in our example and mitigating scaling errors. In our city area example, the Bray-Curtis distance between the areas in Rows 3 and 4 is 0.7563, indicating that Row 3 is considered closer

| Entity | Property | Unit |
|---|---|---|
| City | 684.4 | km$^2$ |
| | 115 | |
| | 87,430,000 | m$^2$ |
| | 630,200,000 | |

**Table 2: Area of a few Canadian cities**

to Row 4 than Row 1, as the distance between Rows 1 and 3 is 0.9999. Our experiments in Section 5.5 confirm that this adaptation results in more accurate unit-based clustering with the k-means algorithm. Additionally, the Bray-Curtis distance is a metric, ensuring smooth integration within clustering algorithms.

**Clustering.** We employ a clustering approach based on the k-means algorithm, with k set to the number of units, to partition a mixed-unit column into sub-columns. Initially, we assume the number of units is known, a presumption we relax in Section 4.3. Assuming adequate differentiation between units within the column, each resulting sub-column should predominantly contain values associated with the same semantic type and unit. Next, we discuss the annotation process for each sub-column.

*4.2.2 Annotating Sub-Models.* In our model generation, a query column is partitioned into sub-columns, and each sub-column is expected to map to an atomic semantic type and unit. This mapping is performed using a single-unit annotation method; in our case, this is done using SAND, as discussed in Section 4.1. However, there are two issues in using a single-unit annotation process. First, while the knowledge graph may be complete in terms of the semantic types covered, it is less likely to include all possible units for each property or sufficient samples for each unit. Thus, the knowledge graph may not have the exact unit that matches a query sub-column. To address this, we compile a set of possible units and conversion rates between all convertible units. When comparing a candidate column from the knowledge graph to a query column, we expand the knowledge graph column with all other applicable units. This type expansion is also applied to data samples using our conversion tables, meaning each new unit will have data samples. For example, if the knowledge graph has human height in centimeters, the data samples in centimeters are mapped to meters, feet, etc.

Another issue is the cost function in SAND, which is defined as the sum of the edge weights in a mapping, with the edge weight determined by the absolute difference between matching quantities. We replace this function with one that provides a more accurate cost estimate when dealing with multiple units, as discussed in the next subsection.

*4.2.3* **Aggregating Sub-Model Annotations.** Our MixedSAND model treats sub-column type predictions as possible candidates for the entire column. The cost of a prediction for the entire column is defined as the cumulative costs of sub-column predictions. However, this cost model is only meaningful if the costs across different sub-columns are comparable. With the default edge weight in SAND, defined as the absolute difference between the quantities connected by the edge, these costs are not comparable across different units.

We modify the edge weights so that the weight of an edge connecting two quantities u and v is defined again using Bray-Curtis distance (Eq. 2). This new edge weight ensures the cost is normalized based on the scale of the quantities matched, making the mappings costs across different sub-columns and candidate types comparable.

### 4.3 Relaxing the Assumption on Unit Count

We now relax the assumption from the previous section that the number of units in a query column is known. Detecting the number of units solely based on the distribution of the values is not straightforward especially when the ranges of values for different units overlap. Each choice of $k$ yields a model of data with an associated cost. Our hypothesis posits that *a correct model should yield a better mapping of the query column and result in the least cost.*

Based on this hypothesis, we vary the number of units $k$ from one to a maximum and estimate the cost of the mapping under each value of $k$. For example, for $k = 2$, our query column is divided into two sub-columns using k-means clustering. With each sub-column assumed to contain only one unit, we proceed with the annotation process individually for each. This results in a final annotation with an associated cost value for each sub-column. The cumulative cost of the two sub-column mappings gives the cost for having two units. We expect the range of possible values for $k$ to be small, and that the optimal number of partitions, where the cost value is minimized, signifies the ideal partitioning of the column into its constituent units.

## 5 EVALUATION

We evaluated our model's performance on diverse datasets with varying parameters, such as data reflectivity, column size and unit counts, and compared it to state-of-the-art baselines.

### 5.1 Experimental Setup

We employ two diverse datasets in our evaluation:

- Mixed-unit datasets: As detailed in Section 3, these are mixed-unit datasets derived from the Wikidata knowledge graph. We generated three variations–Easy, Medium, and Hard–to assess the model's robustness across different levels of unit overlap and reflectivity.
- WDC dataset: This single-unit dataset, used in the evaluation of SAND [24], is a subset of the WDC table corpus. We employ the same subset, enabling a direct comparison with the state-of-the-art method on a standardized benchmark.

More specialized datasets, such as VUCD (Section 5.2), and baselines are introduced in relevant sections.

### 5.2 Detecting the Number of Units

*Single-unit vs. multi-unit columns.* This evaluation was conducted on both our Mixed-unit dataset and the single-unit WDC dataset. To provide context for our model's performance, we used Kernel Density Estimation (KDE) [22], a common method for identifying multimodal distributions, as a baseline. Given that a mixed-unit column contains data from different units (e.g., weight in pounds and kilograms), and because of the differences in range and scale between units, the overall distribution of the column is likely to appear as if composed of two or more distinct distributions. As one can see in Figure 5, this phenomenon manifests as multimodality, where the distribution exhibits multiple peaks or modes. KDE, a non-parametric method for estimating probability density functions, helps assess whether a column's distribution is multimodal [4, 23]. The presence of peaks in a KDE plot suggests a mixed-unit column, while a smooth, unimodal distribution is more characteristic of a single-unit column. We expect the KDE baseline to perform well in cases where units are clearly separated but anticipate challenges in scenarios with high reflectivity, where overlapping units might obscure multimodality.

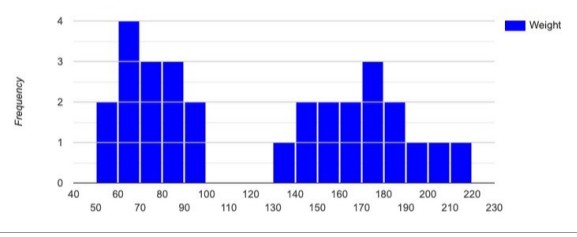

**Figure 5: Distribution of a mixed-unit column**

As shown in Table 3, our model consistently outperformed the KDE-based baseline across all datasets, demonstrating its ability to discern subtle unit patterns more effectively. Both MixedSAND and KDE exhibit relatively lower accuracy on the hard dataset due to high reflectivity, where significant overlap between units makes it more challenging to identify mixed unit columns.

| Dataset | **Hard** | **Medium** | **Easy** | **WDC** |
|---|---|---|---|---|
| Number of Columns | 200 | 200 | 200 | 69 |
| MixedSAND Accuracy | 0.58 | 0.715 | 0.765 | 0.696 |
| KDE Accuracy | 0.50 | 0.54 | 0.55 | 0.652 |

**Table 3: Accuracy of MixedSAND (our method) in identifying multi-unit columns, compared to a KDE-based baseline**

*Determining the number of units.* To evaluate our model's accuracy in counting distinct units within mixed-unit columns, we constructed the Varying Unit Count Dataset (VUCD) with 300 distinct columns. The dataset was generated based on Wikidata, following the mixed-unit dataset generation process outlined in Section 3.2. We categorized the columns based on the number of units they contained (2, 3, 4, or 5), and ensured that the dataset included an equal

number of columns for each unit category. MixedSAND achieved an accuracy of 0.516 in determining the number of units, compared to KDE, which had an accuracy of 0.193. This accuracy is for the entire VUCD, which includes columns with 2 to 5 units. The substantial performance difference likely stems from KDE's reliance on density estimation, which becomes less reliable when distributions are not well-separated, leading to diminished accuracy in such cases.

## 5.3 End-to-End Type Detection Performance

For an end-to-end evaluation, we processed columns without prior knowledge of whether they contained mixed units, fed them into each model, and evaluated performance based on the predicted semantic types. We compared our method against both an LLM-based approach using GPT-4o-mini and the SAND method as baselines. The LLM was tasked with identifying the semantic types of numeric columns in a few-shot setting [5]. While LLM-based solutions have shown strong performace on textual data [11], we sought to evaluate their performance on numeric data, especially in cases with mixed units.

Table 4 presents top-n accuracy, measuring the fraction of annotated columns where the correct semantic type appeared in the top-n predictions, for n=1, 3 and 5. Our model consistently outperforms SAND and GPT-4o-mini across all mixed-unit datasets. The performance gap between our model and SAND is most pronounced in the Easy dataset compared to Medium and Hard. This is because the Easy dataset primarily contains low-reflectivity data, where different units in the same column exhibit minimal overlap. SAND struggles with this separation, as it attempts to annotate all column values at once. However, in the Hard dataset, where there is significant overlap between units, SAND treats the column as a single unit and performs better. It should be noted that the higher degree of overlap in the Hard and Medium datasets makes unit separation more challenging, leading to a slight decrease in accuracy compared to the Easy dataset.

| Dataset | Method | Top-n Accuracy | | |
|---|---|---|---|---|
| | | n=1 | n=3 | n=5 |
| Easy | SAND | 0.195 | 0.36 | 0.53 |
| | GPT-4o-mini | 0.365 | 0.555 | 0.625 |
| | MixedSAND | 0.485 | 0.75 | 0.825 |
| Medium | SAND | 0.22 | 0.455 | 0.65 |
| | GPT-4o-mini | 0.26 | 0.415 | 0.52 |
| | MixedSAND | 0.335 | 0.64 | 0.80 |
| Hard | SAND | 0.275 | 0.455 | 0.665 |
| | GPT-4o-mini | 0.265 | 0.45 | 0.55 |
| | MixedSAND | 0.33 | 0.60 | 0.775 |

**Table 4: Top-n accuracy of MixedSAND vs. SAND and GPT 4o-mini on semantic labeling of the Easy/Medium/Hard datasets**

On the WDC dataset, which consists exclusively of single-unit columns, our model performs comparably to both the LLM-based approach and SAND, as shown in Table 5.

| Method | Top-n Accuracy | | |
|---|---|---|---|
| | n=1 | n=3 | n=5 |
| KS-test | 0.069 | 0.129 | 0.277 |
| SAND | 0.116 | 0.232 | 0.42 |
| GPT 4o-mini | 0.13 | 0.26 | 0.29 |
| MixedSAND | 0.101 | 0.26 | 0.405 |

**Table 5: Top-n accuracy of MixedSAND vs. SAND, KS-test, and GPT 4o-mini on semantic labeling of the WDC dataset**

## 5.4 Performance on Clustered Columns

Sometimes, clusters within a mixed column are known from earlier preprocessing steps. For examples, when data is integrated from multiple sources, each source may be treated as a cluster. This arises particularly after table integration tasks such as table union and stitching [12, 16], where the columns being integrated may share the same type and property but not necessarily the same unit. In such cases, our pipeline can be reduced to two steps: Annotating Sub-Models (Section 4.2.2) and Aggregating Sub-Model Annotations (Section 4.2.3).

To assess the performance of our model in these scenarios, we constructed a real dataset comprising 23 table pairs, with each pair containing two numeric columns to be integrated. The numeric columns in each pair shared the same type and property but differed in units. The dataset was sourced from Wikitables, which includes 1.9 million tables extracted from Wikipedia. We systematically identified numeric columns representing the same property and manually annotated a subset to determine their type and unit. From this, we selected column pairs with identical types and properties but different units to create the dataset for this evaluation.

As shown in Table 6, our model achieved higher accuracy in this specific task compared to the scenarios where data clusters are unknown. This is because, in our evaluation of mixed-unit and single-unit columns (§5.3), the model must first determine whether a column is mixed-unit before proceeding to the annotation step, introducing potential errors at each stage.

| Top-n Accuracy | | |
|---|---|---|
| n=1 | n=3 | n=5 |
| 0.65 | 0.78 | 0.91 |

**Table 6: Post-Clustering Annotation Performance of the Model on the Wikitables Dataset**

## 5.5 Impact of Relative Difference

The impact of using relative difference (or Bray-Curtis distance) instead of absolute distance was evaluated in two key tasks: determining the number of units within mixed-unit columns (multi-class classification) and distinguishing between single-unit and mixed-unit columns (binary classification). The former was assessed using the VUCD dataset, which contains mixed-unit columns with varying unit counts, while the latter was evaluated on both our mixed-unit dataset and the WDC.

As shown in Table 7, our model consistently achieves superior performance when employing relative difference. As discussed in

---

[5]Details on the LLM-based prompt and examples can be found in Appendix A

Section 4.2.1, mixed-unit columns include values measured on different scales (e.g., kilograms vs. pounds), where absolute differences can be misleading. In contrast, relative difference accounts for these scale variations, allowing for more accurate clustering and unit count determination.

| Method | Accuracy |
|---|---|
| MixedSAND + relative difference | 51.6 |
| MixedSAND + absolute distance | 25 |

**Table 7: Impact of relative distance on clustering accuracy**

When differentiating between single-unit and mixed-unit columns, Table 8 reveals a clear improvement in performance when using relative difference. This improvement can be attributed to the inherent limitations of absolute distance in handling mixed-unit data. Large absolute differences can occur between values of the same unit if their magnitudes are high, leading to separate clusters. Conversely, small absolute differences between values of different units with low magnitudes can result in two distinct units being clustered together. Relative difference mitigates these issues by considering the scale of the values, thus improving the model's ability to correctly distinguish between single-unit and mixed-unit columns.

| | Hard | Medium | Easy | WDC |
|---|---|---|---|---|
| Number of Columns | 200 | 200 | 200 | 69 |
| MixedSAND with relative difference | 0.58 | 0.715 | 0.765 | 0.696 |
| MixedSAND with absolute distance | 0.52 | 0.64 | 0.665 | 0.493 |

**Table 8: Impact of relative difference in distinguishing single-unit vs. mixed-unit columns**

## 5.6 Robustness to Variations in Unit Count

To evaluate the robustness of our method with respect to the number of units within a mixed-unit column, we constructed datasets following the process outlined in Section 3.2. These datasets contained mixed-unit columns with varying unit counts while maintaining other parameters, such as reflectivity, unchanged. We constructed four datasets, each containing columns with a specific number of units ranging from two to five. Each dataset contained 100 columns, each with 40 rows, and the reflectivity distribution of the columns aligned with that of the easy dataset (§3.2).

| # of units | Top-n Accuracy | | |
|---|---|---|---|
| | n=1 | n=3 | n=5 |
| 2 | 0.49 | 0.74 | 0.82 |
| 3 | 0.41 | 0.69 | 0.80 |
| 4 | 0.41 | 0.70 | 0.80 |
| 5 | 0.42 | 0.70 | 0.81 |

**Table 9: Performance of MixedSAND on Columns with Varying Numbers of Units**

As shown in Table 9, our model demonstrates robustness to variations in the number of units within a column. Notably, performance remains consistent across all datasets with 3, 4, or 5 units.

## 6 IMPACT OF QUERY COLUMN SIZE

For this evaluation, we constructed a dataset of single-unit columns derived from real-world numeric data in Wikidata, with each column containing 50 entries. To evaluate the impact of query column size on the model's accuracy, we used random samples from the query columns rather than the entire column.

As illustrated in Figure 6, the top-n accuracy initially increases with the length of the query column. However, beyond a certain point, further increases in the query column length results in a decline in accuracy. When the query column size exceeds that of the candidate column, injective mapping becomes unfeasible, necessitating that the query column be sampled to match the candidate column's length. Additionally, for an injective mapping, the data in the query column must map entirely to the data in the candidate column, and the presence of outliers can clearly lead to inaccurate predictions by the model. In our dataset, the average size of a candidate column was 43, and a drop in performance occurs as the query size approaches or surpasses the size of the candidate column.

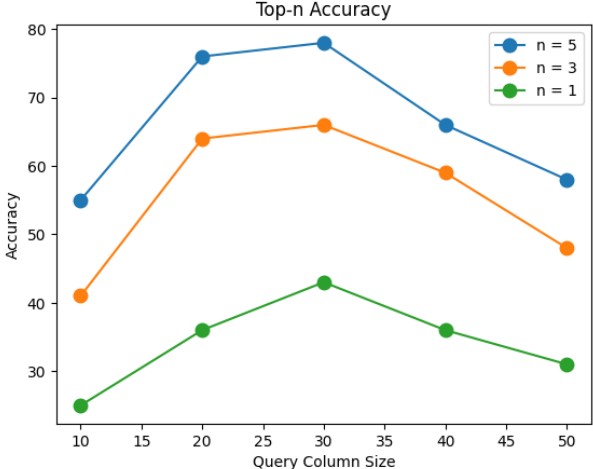

**Figure 6: Accuracy varying query column size**

## 7 CONCLUSIONS

In this paper, we addressed the problem of annotating numerical columns, with a specific focus on mixed-unit columns–an aspect that has been overlooked in previous work. We introduced a multi-unit column annotation benchmark and proposed a new annotation approach that leveraged the SAND model. We evaluated our model on datasets containing mixed-unit columns with varying levels of reflectivity and compared its performance with state-of-the-art methods. The results demonstrated that our method outperforms existing approaches in terms of accuracy.

Future research could explore several promising directions based on our current approach. One avenue is optimizing the process by pre-determining whether a column is mixed-unit before annotation, potentially reducing runtime. Another direction is leveraging the shared context when annotating multiple columns of the same table.

# A DETAILS OF OUR LLM-BASED APPROACH USING GPT-4O-MINI

In this section, we provide detailed information about the LLM-based approach (GPT 4o-mini) used for semantic type detection in the evaluation section. The LLM was prompted to determine the semantic type of a numeric column based on numeric values and their related entity types. Below, we include the prompt used, examples provided to the model, and the evaluation methodology.

## A.1 Prompt Design

The LLM-based method was evaluated using a custom-designed prompt. The task was to identify the most likely semantic type for a numeric column given its entity type. The prompt explicitly mentioned that the numeric column could contain mixed-unit data, but the model was not required to identify the units. Below is the prompt used:

---

I have a dataset with two columns: one contains numeric values, and the other is related to an entity type. The numeric column might contain mixed-unit data, meaning the values could be in different units, but we are not concerned with determining the units. Your task is to determine the most likely semantic type of the numeric column based on the numeric values and the type of entity column they are related to. The semantic type refers to what the numeric values represent in relation to the entity type, such as weight, height, price, duration, or another meaningful concept.
Here are some examples:
1.
Input:
Entity Type: lake
Numeric Column: ['27', '70', '25', '10', '10', '55', '16', ...]
Output: depth
2.
Input:
Entity Type: weapon model
Numeric Column: ['77168.0', '9760.0', '17700.0', '15.8', ...]
Output: mass
3.
Input:
Entity Type: destroyed building or structure
Numeric Column: ['119.5', '85.0', '110.0', '209.0', ...]
Output: length
Now, based on this information, here is the input:
Entity Type: `entity_type`
Numeric Column: `numeric_column`
Please provide up to 5 guesses for the most likely semantic type of the numeric column, without any additional explanation or context.

---

## A.2 Examples Provided to the Model

To improve performance, we included infrequent examples and entity types in the prompt, since the model was already able to correctly predict the semantic type in cases involving frequent types, such as human height and weight. By using more infrequent examples, we aimed to evaluate the model's ability to generalize beyond common cases.

## A.3 Testing Different Prompts

Initially, we tested prompts that did not mention the possibility of mixed-unit columns. In these cases, the model assumed all numeric columns were single-unit and returned lower performance. When we explicitly mentioned that columns might be mixed-unit, the model attempted to annotate the units as well, which was not required and led to reduced accuracy. Therefore, we added the clarification that determining the units was not part of the task, and this resulted in improved performance. The prompt provided above yielded the best overall performance.

## A.4 Evaluation Methodology

For evaluating the LLM-based approach, we followed the same procedure used for evaluating SAND and MixedSAND. The LLM was provided with numeric columns and their corresponding entity types, and the model's predicted semantic types were recorded. As our performance metric, we used top-n accuracy, measuring the fraction of annotated columns where the correct semantic type appeared in the top-n predictions, for n=1, 3 and 5. The performance results of GPT 4o-mini were included in Tables 4 and 5 in the main paper.

## A.5 Conclusion of the LLM Evaluation

The results of the LLM evaluation showed that while GPT 4o-mini could handle simple cases with single-unit columns, it consistently struggled with mixed-unit columns. This confirms the importance of a specialized method for handling mixed-unit numeric data, such as the one proposed in this work.

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
