# OpenReview forum: "MixedSAND: Semantic Annotation of Mixed-unit Numeric Data"
_ACM.org/TheWebConf/2025/Conference — WWW 2025 Oral_

### Official Review · Reviewer_JxDx · 2024-11-19

**Novelty:** 6
**Technical Quality:** 5

**Review:**

The paper addresses an aspect of the problem of connecting tabular data with insufficient metadata to semantic data in a knowledge graph.  More specifically, the problem is to discover which properties in a knowledge graph a given column in a table corresponds to, known as "column type annotation." Even more specifically, the paper describes an approach to this when a) the data is numeric and b) there has been some mixing up of units of measurements within in a column of the table.

Suitable benchmark data for this problem was not available, so one of the contributions is to provide benchmarks of varying difficulty.

The paper presents the MixedSAND approach based on the previously published "SAND" method for tables with a common unit.  MixedSAND starts by clustering column values according to the Bray-Curtis distance, in order to separate values measured using different units.  The SAND approach for annotation is used on the clusters, but with modifications to ensure that the "semantic type" is the same for the whole column, as well as using a suitably adjusted loss function.

Overall, the paper is well written and easy to follow.  I don’t find the approach particularly surprising, but that is a good thing. The experimental results look promising.

I think this would be a good contribution to the track.

Comments:

The paper talks about "semantic types" from the beginning, but I had to look at other papers to find out what is really mean here.  It would be good to give some intuition for what one is trying to detect already in the introduction.

"recent work attempt" → "recent work attempts"
"contribution include" → "contributions include"

Reflectivity: I suggest parameterising theta and rho with the value r.  A definition where a variable occurs on only one side looks really strange.  Maybe by adding r as a subscript to theta and rho.

More seriously: in the given examples, reflectivity works.  But it still comes out of thin air.  What properties are needed for the measure of difficulty here?  Does reflectivity display these properties?  What other measures do?  Why is this a good metric?

"shown to be the state-of-the-art performance" → "shown to have sota performance"

"prone to fail to produce satisfactory results" → "prone to failure…" but better "may not produce satisfactory results"

"Figure 3" write Figure~3 to prevent line break

4.2.2 you describe a unit conversion table.  But do you also have a list of common units for semantic types?   E.g. a "height" will always come with a length unit.  If the "dimension" of units in one column disagree then there is a major problem.

Bibliography: there are some missing capitalisations, e.g. in ChatGPT, SAND, SQL, etc.

**Questions:**

Q1. Can you please clarify why reflectivity is a suitable metric here, and why it is the only one considered?

Q2. If I understood the paper correctly, it aims to clean up information in the case where data in different units has already made it into a consolidated table.  Wouldn’t it be a lot easier to identify the correct unit for a given data source before merging it with others?  So we could assume that the units are consistent within one column of one source?  I agree that this is not always the case, but probably often?
Often, the source of information is also kept, like in your Table 1.  Taking this into account would help a lot with the clustering process.

Q3. There could be useful information from other columns.  E.g., country codes can hint at more likely systems of units.  Also, it seems likely that a given data source either uses mostly imperial or US units, or SI.  Do you have any plans of including this kind of information?

**Reviewer Confidence:**

3: The reviewer is confident but not certain that the evaluation is correct

**Scope:**

4: The work is relevant to the Web and to the track, and is of broad interest to the community

---

### Official Review · Reviewer_i77f · 2024-12-01

**Novelty:** 6
**Technical Quality:** 6

**Review:**

The approach MixedSAND brings identification and (semantic) annotation of numeric data for, as the name suggests, mixed-unit data.

A key example for this is for example weight measured in pounds or kg, and similar for length units.

Pros:

* a clearly important problem for semantic integration

* very good discussion of design choices (also of the dataset construction) and context

* good consideration on the concept of "reflectivity"

* convincing evaluation (albeit with - given the new task - limited comparison)

Cons:

I consider the paper very convincing, if one had to find cons, one could say:

* the complexity of the approach is, I woudl say, appropriate for the task at hand; one could ask the question on technical novelty in terms of techniques - although as said, I think it is fully appropriate

* I may have missed it, but I did not find available resources for reproducing the results with code/data

In total, I find it a very convincing paper solving a specific task that is challenging.

What I am wondering - but this is more for related research - what additional work one could find in the area of (relational) databases in this context.


Detailed comments

- "the adaptability of the methods remain questionable": consider explaining this more

- "(We provide more details on SAND in Section 4.1.)": consider different formatting

**Questions:**

I have no direct questions, though please feel free to reply to any points mentioned.

**Reviewer Confidence:**

3: The reviewer is confident but not certain that the evaluation is correct

**Scope:**

4: The work is relevant to the Web and to the track, and is of broad interest to the community

---

### Official Review · Reviewer_szUT · 2024-12-03

**Novelty:** 4
**Technical Quality:** 5

**Review:**

This paper proposes an approach for semantic annotation of numeric data in tabular formats, specifically targeting scenarios where numerical columns contain mixed-unit data (e.g., heights in both meters and feet). The proposed approach is based on three stages (a) model generation: clustering data to separate values into sub-models corresponding to individual units, (b) type annotation: annotating each cluster using knowledge graph mappings, (c) aggregation: Combining sub-model annotations into unified column annotations.

Pros:
- experimental results demonstrate better performance of MixedSAND compared to the baselines tested in the paper (KS-test, SAND, GPT-4o-mini), particularly on mixed-unit datasets.
- the authors introduce a dataset for mixed-unit numeric column annotation with easy, medium, and hard variations that will be a useful contribution to the community and future research on semantic annotation of numeric data in tabular formats.
- The addressed problem is quite relevant to the real-world challenges in data lakes because most real world enterprise data have similar challenges.

Cons:
- The paper lacks a discussion on the limitations of the current approach and an error analysis of the results.
- Further explanations and insights or intuitions on the behaviours seen on the impact of query column size and the robustness to variations in unit count could help the readers.

**Questions:**

- Have you analyzed how sensitive the final performance is to errors in the initial clustering step? Would sub-optimal clustering significantly degrade the final annotations?
- What are the challenges the authors forsee in adapting the proposed approach to streaming or dynamic datasets where new units or data distributions may emerge over time?

**Reviewer Confidence:**

3: The reviewer is confident but not certain that the evaluation is correct

**Scope:**

3: The work is somewhat relevant to the Web and to the track, and is of narrow interest to a sub-community

---

### Official Review · Reviewer_3bv7 · 2024-12-04

**Novelty:** 5
**Technical Quality:** 6

**Review:**

The authors propose a framework for investigating the conditions for effectively annotating mixed-unit numeric
data and introduce a benchmark for an annotation task. It also proposes an algorithm that reliably detects semantic types
and links them to KGs. Their results show that their framework generated comparable results with the baselines.

Pros:
1. The proposed model can handle mixed-unit numeric data, whereas SAND is designed for single-unit columns.
2. A benchmark dataset has been proposed
3. The results show that the proposed model gives comparable results with the baselines

Cons:
Results on the existing dataset as shown in Table 5, the performance of the baseline model SAND is much better that MixedSAND model. Why is it so? Does it indicate that the performance of the MixedSAND model drops when it comes to single-unit columns? What is the proportion of occurrences the single unit column and mixed unit data in real-world scenarios?

**Questions:**

1. As the model is largely dependent on the KG information (mostly Wikidata), how can the model be generalised to other KGs which are domain specific?
2. How do we interpret the performance based on Table 6, as no baseline models have been discussed so far?
3. why are the results better in SAND than Mixed SAND in Table 5 and why mixed sand is better in Table 4?

**Reviewer Confidence:**

3: The reviewer is confident but not certain that the evaluation is correct

**Scope:**

3: The work is somewhat relevant to the Web and to the track, and is of narrow interest to a sub-community